# A human respiratory tract-associated bacterium with an extremely small genome

Kazumasa Fukuda [1✉], Kei Yamasaki[2], Yoshitoshi Ogura[3], Toshinori Kawanami[2], Hiroaki Ikegami[2], Shingo Noguchi[2], Kentarou Akata[2], Keisuke Katsura[4], Kazuhiro Yatera[2], Hiroshi Mukae[5], Tetsuya Hayashi [6✉] & Hatsumi Taniguchi[1]

Recent advances in culture-independent microbiological analyses have greatly expanded our understanding of the diversity of unculturable microbes. However, human pathogenic bacteria differing significantly from known taxa have rarely been discovered. Here, we present the complete genome sequence of an uncultured bacterium detected in human respiratory tract named IOLA, which was determined by developing a protocol to selectively amplify extremely AT-rich genomes. The IOLA genome is 303,838 bp in size with a 20.7% GC content, making it the smallest and most AT-rich genome among known human-associated bacterial genomes to our best knowledge and comparable to those of insect endosymbionts. While IOLA belongs to order Rickettsiales (mostly intracellular parasites), the gene content suggests an epicellular parasitic lifestyle. Surveillance of clinical samples provides evidence that IOLA can be predominantly detected in patients with respiratory bacterial infections and can persist for at least 15 months in the respiratory tract, suggesting that IOLA is a human respiratory tract-associated bacterium.

[1] Department of Microbiology, University of Occupational and Environmental Health, Kitakyushu, Fukuoka, Japan. [2] Department of Respiratory Medicine, University of Occupational and Environmental Health, Kitakyushu, Fukuoka, Japan. [3] Division of Microbiology, Department of Infectious Medicine, Kurume University School of Medicine, Kurume, Fukuoka, Japan. [4] Frontier Science Research Center, University of Miyazaki, Miyazaki, Japan. [5] Department of Respiratory Medicine, Unit of Translational Medicine, Nagasaki University Graduate School of Biomedical Sciences, Nagasaki, Japan. [6] Department of Bacteriology, Faculty of Medical Sciences, Kyushu University, Fukuoka, Japan. ✉email: kfukuda@med.uoeh-u.ac.jp; thayash@bact.med.kyushu-u.ac.jp

Recent advances in culture-independent microbiological analyses due to technical innovation in DNA sequencing technologies[1] have greatly expanded our understanding of the vast diversity of microbes in various environments, which was never achieved by culture-based methods[2]. Culture-independent methods have been applied to various clinical specimens to assess pathogenic micro-organisms, as well as commensal microbes inhabiting our bodies. However, human pathogenic bacteria that significantly differ from previously known taxa have rarely been discovered from human-derived clinical samples, in sharp contrast to the findings in the field of environmental microbiology[3–8].

We previously identified an unusual 16S rRNA gene sequence in bronchoalveolar lavage fluid (BALF) specimens from patients with respiratory tract disorders, during the assessment of microbiota of various clinical specimens by the 16S rRNA gene clone library method[9]. Although attempts to isolate the micro-organism were unsuccessful, we temporarily named it Infectious Organism Lurking in human Airways (IOLA) and reported several of its characteristics. The size of IOLA cells was estimated to be approximately 1 μm with a spherical shape, and sequencing analysis of a genomic fragment suggested an extremely low GC content (approximately 20%)[9]. Although only a few 16S rRNA sequences similar to that of IOLA (>96% identity) have been deposited in the NCBI database (https://www.ncbi.nlm.nih.gov), they were all identified in specimens from patients with respiratory tract disorders, including the first registered IOLA-like 16S rRNA sequence detected in a BALF sample from a cystic fibrosis patient[10]. These data suggest that IOLA may be a previously unknown human respiratory pathogen. Recently, IOLA-like 16S sequences were predominantly detected in the tracheal samples of a swine transplanted with bioengineered lung[11], suggesting that IOLA may be an agent of zoonotic infection.

Here, we report the whole genome sequence (WGS) of IOLA reconstructed from the metagenomic sequences of a BALF sample by developing a protocol to selectively amplify extremely AT-rich genomes. Analyses of the WGS revealed many intriguing genetic features of IOLA. For example, it has an extremely small genome size among known human-associated bacterial genomes, which is equivalent to the sizes of known insect symbiont genomes, and its unique gene content indicates a parasitic lifestyle distinct from that of insect symbionts. The results of a survey of respiratory disease patients using IOLA-specific PCR and an analysis of multiple samples obtained in the survey are also presented, which suggest that IOLA can inhabit and persist in the respiratory tract of patients with respiratory disorders.

## Results

**Determination of the WGS of IOLA.** To reconstruct the WGS of IOLA, we established a protocol for metagenomic DNA preparation enriched with IOLA DNA (Fig. 1). In this protocol, human-derived cells/tissues and larger bacterial cells in a BALF specimen were prefiltered, and smaller bacterial cells were collected with a membrane filter and washed with PBS to remove free DNA. Total DNA was then extracted and subjected to whole-genome amplification (WGA). In the WGA process, DNA was denatured at an optimized temperature (80 °C) to selectively denature the highly AT-rich DNA. Using this protocol, we successfully created an IOLA-enriched DNA preparation from a specimen from sample KY-405 (a sample from patient A in our previous study[9]). In this sample, the IOLA DNA was markedly enriched from 11.5 to 82.7% as estimated by 16S rRNA clone library analyses of the samples with or without this protocol (Fig. 1). We prepared a paired-end and mate-pair library from the amplified DNA sample for Illumina sequencing. After quality trimming and filtering of human-derived reads, the Illumina

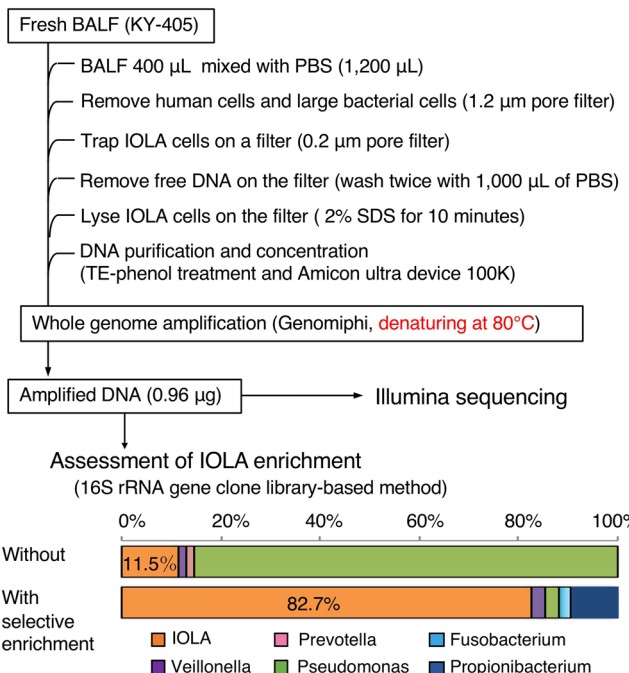

**Fig. 1 Protocol for selective amplification of the IOLA genome.** The processes of DNA extraction and selective amplification of IOLA genomic DNA are shown. The proportions of IOLA DNA in the DNA preparations obtained with or without selective amplification were estimated by the 16S rRNA clone library-based method (61 and 75 clones were analyzed, respectively).

reads were assembled using the SPAdes[12] and Platanus[13] assemblers. We selected scaffolds of >1000 bp in length and with a low GC content (<25%). From this scaffold sets, the scaffolds which were found to be derived from known organisms or to be low coverage (less than 5×) were removed, yielding 15 and 25 scaffolds from SPAdes and Platanus assemblies, respectively (Supplementary Fig. 1). The 40 scaffolds were aligned and merged into nine scaffolds. By closing the gaps ($n = 9$) and ambiguous regions ($n = 18$) in these scaffolds by PCR and Sanger sequencing (see Supplementary Fig. 1 for details), we obtained a circular 303,838-bp chromosome of IOLA KY-405 with an average GC content of 20.7% (Fig. 2a).

To verify that the final assembly is not missing any sequence of IOLA, we analyzed, all SPAdes and Platanus scaffolds that were not used for the final assembling step (see Supplementary Fig. 2 for details), and found that most of them (89.2% of the SPAdes scaffolds and 92.7% of the Platanas scaffolds) showed high sequence similarities to the sequences of human or other known organisms (E-value; <1.00E−100) and/or had high GC contents (>50%). Among the remaining scaffolds, several scaffolds were found to correspond to the gap regions of IOLA assembly. They were smaller than 1 kb ($n = 6$) or had a GC content slightly higher than 25% ($n = 1$). Other scaffolds were clearly distinguished from the scaffolds used in the final assembly based on GC content and sequencing coverage (depth). Thus, it is very unlikely that the final assembly is missing some sequences of IOLA.

**General genomic features of IOLA.** The IOLA chromosome is among the smallest completely sequenced bacterial genomes, comparable to the genomes of many insect endosymbionts and much smaller than those of *Mycoplasma* species (larger than 564 kb), which have the smallest genomes among culturable bacteria (Supplementary Table 1). The IOLA chromosome contains 310

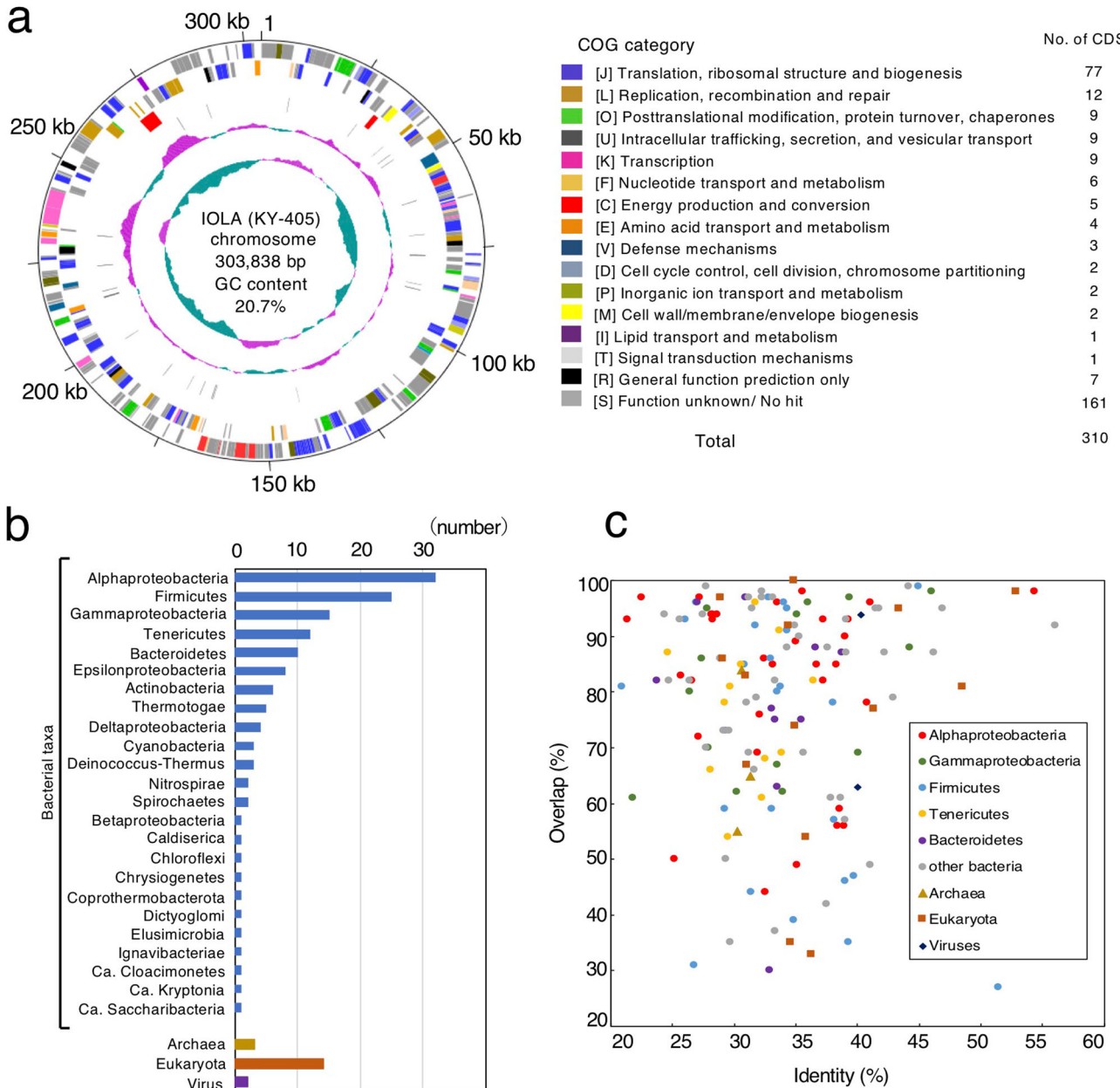

**Fig. 2 CDSs in the IOLA genome and their homologies with known proteins. a** Circular representation of the IOLA genome and COG classification of the CDSs. Circles from outside to inside: 1, scale in kb; 2, forward-strand CDSs; 3, reverse-strand CDSs; 4, rRNA genes; 5, tRNA genes; 6, GC content (graphic scale; upper: 0.3, center: 0.2, lower: 0.1); and 7, GC skew (graphic scale; upper: 0.25, center: 0.05, lower: −0.15). The window size and the step size in circles 6 and 7 are 10,000 bp and 1000 bp, respectively. **b** Taxonomic affiliations of the top-hit proteins in the blastp search of each IOLA CDS. **c** Sequence identity (%) and overlap (%) of IOLA CDSs with the blastp top-hit proteins. Proteins from the top five bacterial taxa, other bacteria (bacterial taxa with proportions less than 5%), archaea, eukaryotes, and viruses are indicated by different colors. Eleven of the 14 eukaryote top-hit CDSs had the top hits to the proteins of mitochondria and chloroplasts.

protein-coding sequences (CDSs) and 34 tRNA genes (Fig. 2a and Supplementary Data 1). Although the 5S rRNA gene was not identified, the 16S and 23S rRNA genes (one copy each) were identified (Supplementary Table 2). The 23S rRNA gene contained three introns; two were homing endonuclease-encoding introns, and one was a group I catalytic intron encoding a self-splicing ribozyme (Supplementary Fig. 3). An intron-encoded homing endonuclease in the 23S rRNA gene has been detected in only a very limited number of bacterial species, such as *Coxiella burnetii*, *Simkania negevensis*, *Thermotoga* species, and *Thermosynechoccus elongates*[14]. Moreover, the presence of multiple homing endonuclease-encoding introns in the 23S rRNA gene is

extremely rare, and has been found in only a few bacteria belonging to the candidate phyla radiation (CPR)[5].

**Phylogenetic placement of IOLA.** Analysis of the IOLA 16S rRNA gene sequence using the Ribosomal Database Project (RDP) Classifier[15] revealed that although IOLA was classified into domain Bacteria with 96% accuracy, it was categorized as an unclassified bacterium at the phylum level, even when using a reduced confidence threshold (60%). Upon homology search using the NCBI databases, the IOLA sequence showed up to 73% identity with known bacterial sequences (except for IOLA-related sequences) (Supplementary Table 3).

In the analysis of CDSs, 153 (49%) of the 310 CDSs identified in IOLA showed no clearly recognizable sequence homology to known proteins, even with lower thresholds, and top-hit sequences to the remaining 157 CDSs were distributed in a great variety of taxa, including 3, 14, and 2 CDSs having a top hit to archaeal, eukaryote (mostly mitochondrial or chloroplastic), and viral proteins, respectively (Fig. 2b, c). However, among the 138 CDSs having a top hit to bacterial proteins (from 24 taxa), the highest proportion (32 CDSs) had a top hit to proteins from Alphaproteobacteria (Fig. 2b).

To clarify the phylogenetic position of IOLA, we performed a phylogenetic analysis based on 16 ribosomal protein sequences[3] of IOLA and sequences from 122 bacteria representing 56 phyla (29 named and 27 candidate phyla) (listed in Supplementary Data 2). In the phylogenetic tree obtained (Supplementary Fig. 4a), IOLA formed a long branch in phylum Proteobacteria. The phylogenetic tree resulting from 16S and 23S rRNA gene sequence-based analysis also showed a similar topology, placing IOLA in phylum Proteobacteria (Supplementary Fig. 4b).

To clarify the position of IOLA at the class level, we constructed a ribosomal protein-based tree using 451 species belonging to phylum Proteobacteria (listed in Supplementary Data 2). In this tree, IOLA branched from class Alphaproteobacteria (Fig. 3). The branch length of IOLA was much longer than those of other very-low-GC bacteria with a small genome, such as Zinderia (14.5%), Buchnera (20%), Portiera (24%), and Riesia (25%). Further ribosomal protein-based phylogenetic

analysis focusing on Alphaproteobacteria (157 species; Supplementary Data 2) revealed that IOLA formed a unique branch that separated early from family Anaplasmataceae in order Rickettsiales (Fig. 4a). A similar phylogenetic position was also observed in the tree constructed using a set of 12 proteins encoded by other housekeeping genes, which were confidently identified in the IOLA genome (Fig. 4b).

These results suggested that IOLA is a unique lineage belonging to order Rickettsiales. However, because of the very long distance of this bacterium from known lineages, a phylogenetic artifact called "long branch attraction (LBA)"[16] may have occurred in these phylogenetic analyses. Therefore, to further confirm the phylogenetic position of IOLA, 16 ribosomal protein-based and 12 housekeeping protein-based trees were reconstructed using Block Mapping and Gathering with Entropy (BMGE)[17], and the posterior mean site frequency (PMSF)[18] model instead of Gblocks[19] and the LG+F+R10 model, respectively (Supplementary Fig. 5). In both trees, IOLA was placed at the same position as in the original trees. Collectively, the data that we obtained indicate that IOLA is a long branching lineage that is separated from family Anaplasmataceae in order Rickettsiales.

**Biological features inferred by WGS analysis**. The 157 functionally predicted CDSs included 38 ribosomal protein genes, 20 aminoacyl-tRNA synthase genes for all standard amino acids, five translation factors, and several genes associated with the Clusters

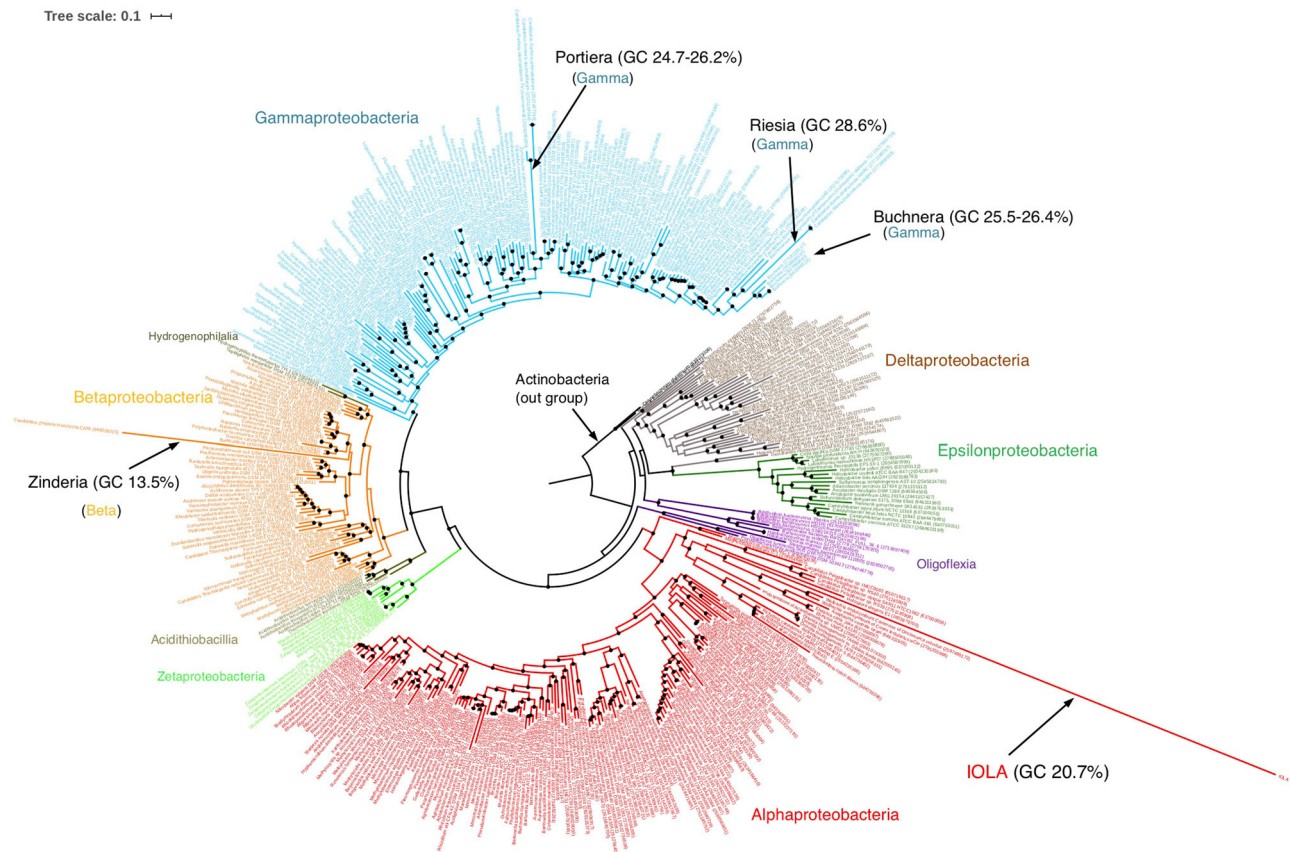

**Fig. 3 Maximum likelihood phylogenetic tree based on the concatenated sequences of 16 ribosomal proteins of Proteobacteria.** The tree includes 451 species belonging to phylum Proteobacteria, two species of Actinobacteria (used as outgroups) and IOLA (listed in Supplemental Data 2). A total of 454 sets of the 16 ribosomal proteins (RpL2, 3, 4, 5, 6, 14, 15, 16, 18, 22, and 24 and RpS3, 8, 10, 17, and 19) were used. Species belonging to the same class are presented in the same color. The tree was constructed using the 1895-position alignment by the maximum likelihood method with the LG + R10 model and 1000 ultrafast bootstrap replicates. The scale bar indicates substitutions per site. Black dots indicate divergence episodes with bootstrap values greater than 80%. Four species that have a small genome (<1.0 Mb) with a low GC content (<30%), such as IOLA, are indicated.

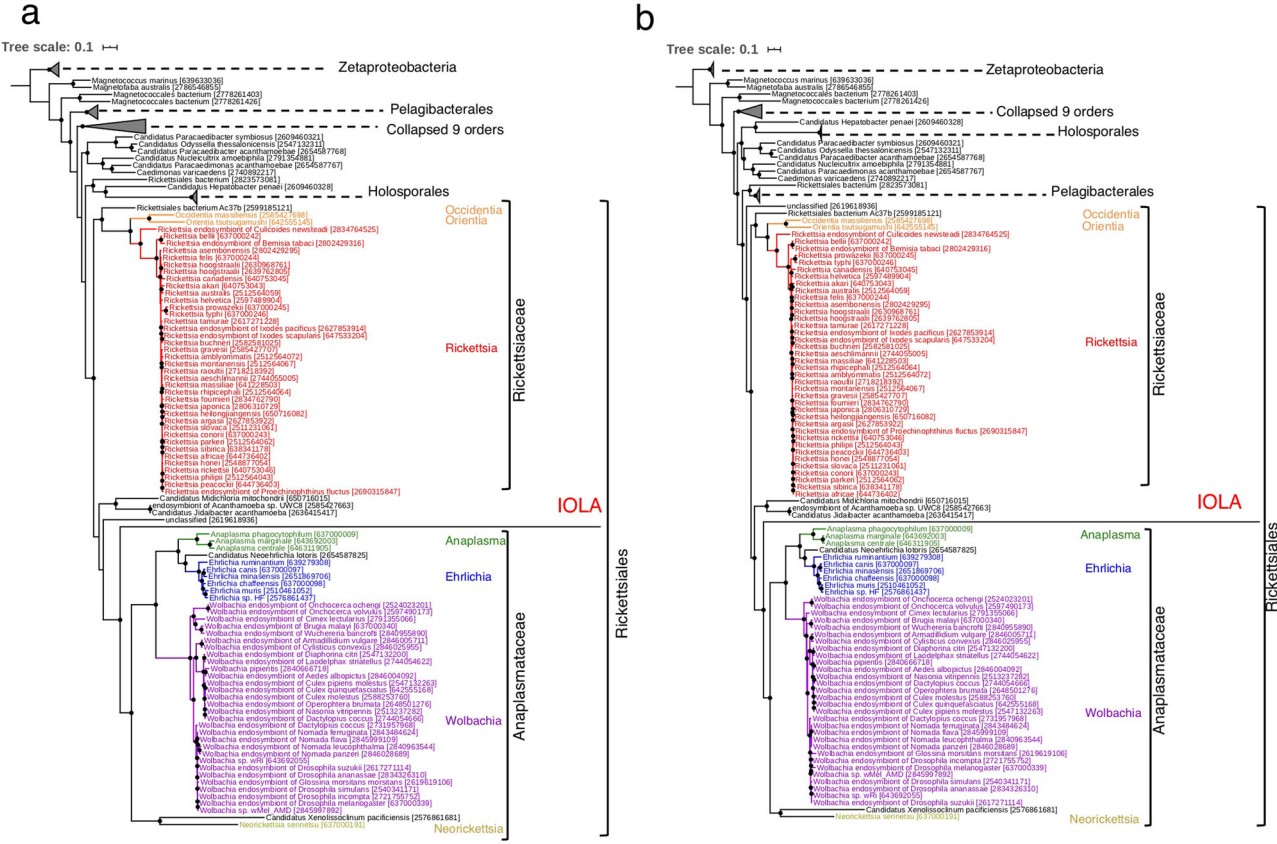

**Fig. 4 Maximum likelihood phylogenetic trees based on the concatenated sequences of ribosomal and other housekeeping proteins of class Alphaproteobacteria. a** The tree includes 157 species belonging to class Alphaproteobacteria, 7 species of Zeta-proteobacteria (used as outgroups) and IOLA (listed in Supplemental Data 2). A total of 165 sets of the 16 ribosomal proteins were used (the same set as used in Fig. 3). **b** The 12 housekeeping proteins (SecA, SecY, TsaD, rpoA, RecA, DnaK, GroEL, GyrA, GyrB, PheS, SerS, and LeuS) of the 165 bacterial species analyzed in **a** were used. Both trees were constructed by the maximum likelihood method with the LG+F+R10 model and 1000 ultrafast bootstrap replicates and either the 1872-position (**a**) or 5048-position (**b**) alignment. Black dots indicate divergence episodes with bootstrap values greater than 80%. The scale bar indicates substitutions per site. In both trees, different genera belonging to order Rickettsiales are presented in different colors.

of Orthologous Groups (COG) functional categories[20] "translation, ribosomal structure and biogenesis" (Fig. 2a and Supplementary Data 1). A set of genes for replication and transcription, such as genes encoding subunits of DNA polymerase, ligase, and RNA polymerase, were also identified, although some genes that are generally essential for these functions were not identified, probably due to their low sequence similarities to the orthologs of other bacteria. Based on COG assignment, no genes categorized into "RNA processing and modification", "carbohydrate transport and metabolism", "coenzyme transport and metabolism", "cell motility", "cytoskeleton", "mobilome", or "secondary metabolite biosynthesis" were identified in IOLA. For "nucleotide transport and metabolism", although genes for the de novo synthesis pathways of purine and pyrimidine were not found, several genes for their interconversion and salvage pathway components, such as adenylate kinase, nucleoside-diphosphate kinases, ribonucleotide reductase, and nucleoside 2′-deoxyribosyltransferase, were identified. In addition, IOLA has an ATP/ADP translocase. This protein may transport not only ATP but also other nucleotides, as shown for several translocases from *Chlamydia* and *Rickettsia* species[21,22].

Genes for the biogenesis of cell wall (e.g., genes for the biosynthesis of peptidoglycan and lipopolysaccharide) that are present in typical bacteria were also not identified, suggesting that IOLA lacks cell wall. Among the lipid biosynthesis genes, only a cardiolipin synthase (cardiolipin is found in the membranes of

most bacteria) gene was identified. The results of SOSUI analysis[23] revealed that 88 IOLA CDSs were predicted to contain at least one transmembrane helix (Supplementary Data 1), while 69 of the 88 CDSs showed no recognizable similarity to known proteins. Although several genes for protein secretion system components, such as preprotein translocase subunits (SecY and SecA) and signal recognition particle GTPases (Ffh and FtsY), were identified, no signal peptides were predicted in any IOLA CDSs by PSORTb[24]. Despite the apparent lack of the ability to synthesize amino acids, carbohydrates, phospholipids, and coenzymes, only a few genes involved in the transport of these compounds were identified, suggesting the possibility that IOLA contains previously unknown systems for the uptake of these compounds.

Regarding "energy production and conversion", IOLA appears to contain no genes for glycolysis and the tricarboxylic acid (TCA) cycle. In addition, no genes for the electron transfer system (genes for NADH: ubiquinone oxidoreductase, succinate dehydrogenase, cytochrome bc complex, and cytochrome c oxidase) were identified. However, genes for three components of F-type ATP synthase and an $H^+$-translocating membrane pyrophosphatase ($H^+$-PPase) were identified. Similar to other IOLA proteins, the $H^+$-PPase sequence of IOLA is highly divergent from known $H^+$-PPase sequences (Supplementary Fig. 6a). However, all key amino acid residues for its function[25,26], except for one of the three Asp residues located in the ion conductance channel, were conserved (Supplementary Fig. 6b).

Phylogenetic analysis with known $H^+$-PPases indicates that the IOLA enzyme belongs to the $K^+$-independent $H^+$-PPase family. The conservation of a Lys residue specific to this $H^+$-PPase family[25,26] supports this notion. As mentioned above, IOLA contains an ATP/ADP translocase gene. In bacteria, ATP/ADP translocases have been found in only a few obligate intracellular bacteria belonging to *Chlamydiales*, *Rickettsiales*, and *Candidatus* Liberibacter[27] and those belonging to candidate phylum TM6, the life cycle of which has been inferred as dependent on eukaryotic hosts[28]. Phylogenetically, the ATP/ADP translocase of IOLA forms a distinct branch neighboring those of Microsporidia (eukaryote) but separated from the cluster of bacterial and chloroplast proteins (Supplementary Fig. 7).

These results strongly suggest the host dependency of IOLA, but any protein motifs that may facilitate eukaryotic host interactions, such as ankyrin repeats, WD40, F-box, leucine-rich repeats, and tetratricopeptide repeats[28], were not detected in IOLA proteins by the motif search using Pfam[29] on the HMMER website (https://www.ebi.ac.uk/Tools/hmmer/). Intriguingly, however, IOLA contained the genes for the type VI protein secretion system (T6SS; five genes were identified)[30].

**Comparison with other bacteria with small genomes**. The results of scatter plot analysis of the 9871 complete bacterial genomes based on genome size and GC content are shown in Fig. 5a. This analysis revealed that IOLA has the smallest genome

with the lowest GC content among the bacteria for which humans are a host. To better understand the biology and lifestyle of IOLA, we compared the gene composition of IOLA with those of other representative host-associated bacteria with small genomes (<1.2 Mb) (Fig. 5b and Supplementary Data 3 and 4). Most of these bacteria are endosymbionts and intracellular or epicellular parasites. Clustering of these bacteria, including IOLA, based on their gene compositions (the proportion of genes belonging to each COG functional category) classified them into five clusters (Fig. 5b). All bacteria in cluster I were insect endosymbionts with extremely small genomes (<0.42 Mb) and a high proportion of genes for amino acid transport/metabolism[31]. Cluster II included insect endosymbionts with larger genomes and bacteria associated with various host organisms. Cluster III consisted of members of family *Anaplasmaceae* (*Anaplasma*, *Wolbachia*, *Ehrlichia*, and *Neorickettsia*) and an arthropod endosymbiont (*Ca.* Profftella). Although the phylogenetic analyses indicated that IOLA was a lineage in order *Rickettsiales*, IOLA was placed in cluster IV with bacteria belonging to phylum *Tenericutes* (*Mycoplasma*, *Ureaplasma*, *Mesoplasma*, *Spiroplasma*, and *Ca.* Phytoplasma), which are known as wall-less bacteria with an epicellular parasitic lifestyle[32]. Cluster V consisted of intracellular parasites associated with various host organisms. *Rickettsia prowazekii*, a member of order *Rickettsiales*, belonged to cluster V. These results suggest that IOLA has a lifestyle similar to those of wall-less epicellular parasites, but different from those of insect endosymbionts and other members of order *Rickettsiales*.

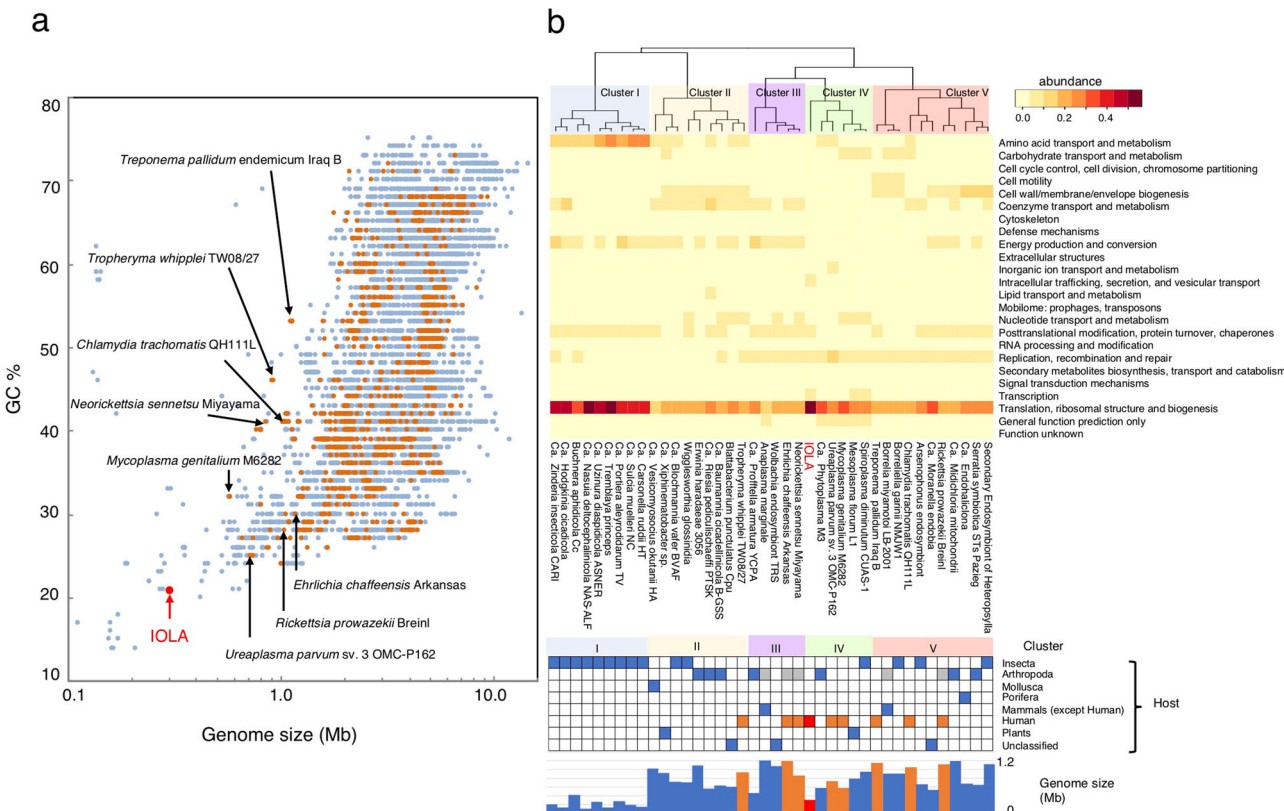

**Fig. 5 Comparison of genomic features of bacteria with extremely small genomes and low GC contents. a** Scatter plot based on the genome sizes and GC contents of 9871 bacteria. Orange dots indicate bacteria with host names registered as "Homo sapiens". IOLA is indicated by a red dot. **b** COG functional category proportions, hosts, and genome sizes of the host-associated bacteria with small genomes are shown. The 39 bacterial species with the smallest genome size in each genus were selected from 265 host-associated bacteria with genome sizes less than 1.2 Mb. The major clusters based on the heatmap of COG functional category proportions are colored. All information, such as the number of genes in each COG functional category, host names, genome sizes, and ecosystems, was collected from the JGI IMG database (http://img.jgi.doe.gov). The arrows in **a** show the human-associated bacteria selected in **b**. The generally known nonhuman vectors or hosts of the human-associated bacteria are shown as gray boxes in "host category".

**Survey of IOLA-positive patients.** To perform a larger-scale survey of IOLA-positive specimens and to efficiently identify them, we developed an IOLA-specific nested PCR system that can detect a few copies of IOLA DNA (Supplementary Fig. 8). Using this system, we screened a total of 523 specimens, including 480 BALF and 43 endotracheal aspirate (EA) specimens, from 501 patients with respiratory diseases, such as pneumonia, bronchitis, and lung abscess. The 523 specimens included 252 BALF and 20 EA specimens from 247 patients, which were previously examined by 16S rRNA clone library-based analysis[9]. Positive results were observed in 14 specimens (12 BALF and two EA specimens from 11 patients; four were from a single patient). Thus, the detection rate of IOLA was 2.7%. Quantification of 16S rRNA gene sequences revealed that the copy numbers of IOLA 16S rDNA in these IOLA-positive samples varied significantly from $(3.5 \pm 2.0) \times 10^3$ to $(2.7 \pm 0.4) \times 10^7$ copies/mL (Supplementary Table 4). The total bacterial 16S rDNA content was over $10^6$ copies/mL in all IOLA-positive samples, indicating lower respiratory tract bacterial infection. In the IOLA-negative lower respiratory samples from patients with interstitial pneumonia (a noninfectious disease), the total 16S rDNA content was below the lower limit of quantification. In microscopic examinations with Papanicolaou, Grocott, and Gram staining, no findings suggesting infections of parasitic eukaryotic organisms were obtained from any of the 14 IOLA-positive specimens. No positive results were obtained in the PCR examination using a primer set specific to the 18S rRNA genes of various nonhuman eukaryotes[33].

Among the 11 positive patients, eight had one or more comorbidities, such as bronchiectasis, allergic bronchopulmonary aspergillosis, bronchial asthma, interstitial pneumonia, diabetes mellitus, rheumatoid arthritis, and hypertension (Supplementary Table 4). In addition, five of the eight patients were being treated with corticosteroids or immunosuppressants. The remaining three patients had no comorbidities. In seven specimens, opportunistic pathogens, such as *Staphylococcus aureus*,

*Pseudomonas aeruginosa*, and *Haemophilus influenzae*, were detected by a standard culture-based examination (Supplementary Table 4).

**Genome diversity of IOLA.** To examine the genetic diversity of IOLA, we amplified 12 segments of the IOLA genome (approximately 1 kb in size, scattered throughout the genome with approximately 25-kb intervals; Supplementary Data 5) by nested PCR from the IOLA-positive samples, because we were unable to apply the protocol shown in Fig. 1 to other specimen due to the lower amounts and qualities (high viscosity and large amounts of DNA from host and other bacteria). Sequencing analysis of the 12 segments (total length: 12,410 bp) revealed that the sequences of four samples obtained from a single patient were identical (KY-41, KY-315, KY-366, and KY-405; KY-405 was used for genome reconstruction) (Fig. 6; more details of this patient are provided in the next section). Other samples showed 2–13 single nucleotide polymorphisms (SNPs; see Supplementary Fig. 9 for all SNP information). Using these SNPs, IOLA samples from different patients were distinguished, enabling the determination of their phylogenetic relationships (Fig. 6) and revealing the notable genetic diversity of circulating IOLA clones. Notably, a highly biased distribution of SNPs was observed; SNPs were identified in only six segments (Fig. 6b). Moreover, in several samples, SNPs accumulated in one or two segments. In particular, 12 of the 13 SNPs in KY-282 were concentrated in a 214-bp region in the PCR50 segment (see Supplementary Fig. 9 for details). Of the 13 SNPs identified in KY-677, 10 were located in two segments: five SNPs each in the PCR175 and PCR250 segments. All SNPs (5/5) in the PCR250 segment and three of the five SNPs in the PCR175 segment of KY-677 introduced nonsynonymous substitutions (Supplementary Fig. 9). Although a possibility of positive selection cannot be excluded, it is more likely that these sequences were introduced by recombination between different clones.

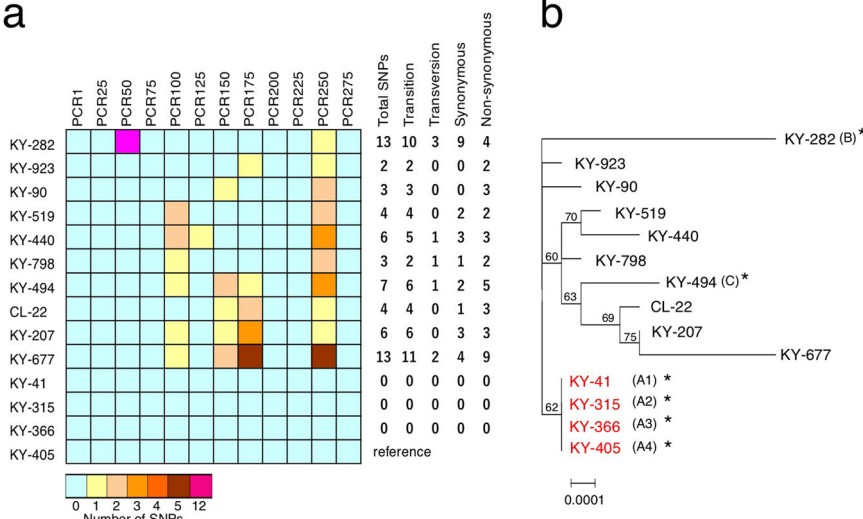

**Fig. 6 Genomic diversity of IOLA. a** Twelve genomic segments (PCR1 to PCR275; approximately 1 kb in size, located in the IOLA genome at approximately 25-kb intervals) were amplified by nested PCR from 14 IOLA-positive clinical specimens and sequenced (see Supplementary Data 5 for the genomic positions of each segment). The numbers of SNPs in each segment (reference: KY-405) and the total number of SNPs and proportions of each SNP type are shown on the right-hand side. **b** An ML tree constructed based on the concatenated sequences of 12 genomic segments (12,410 bp in total). The scale bar indicates substitutions per site. The four specimens indicated in red were obtained from a single patient.

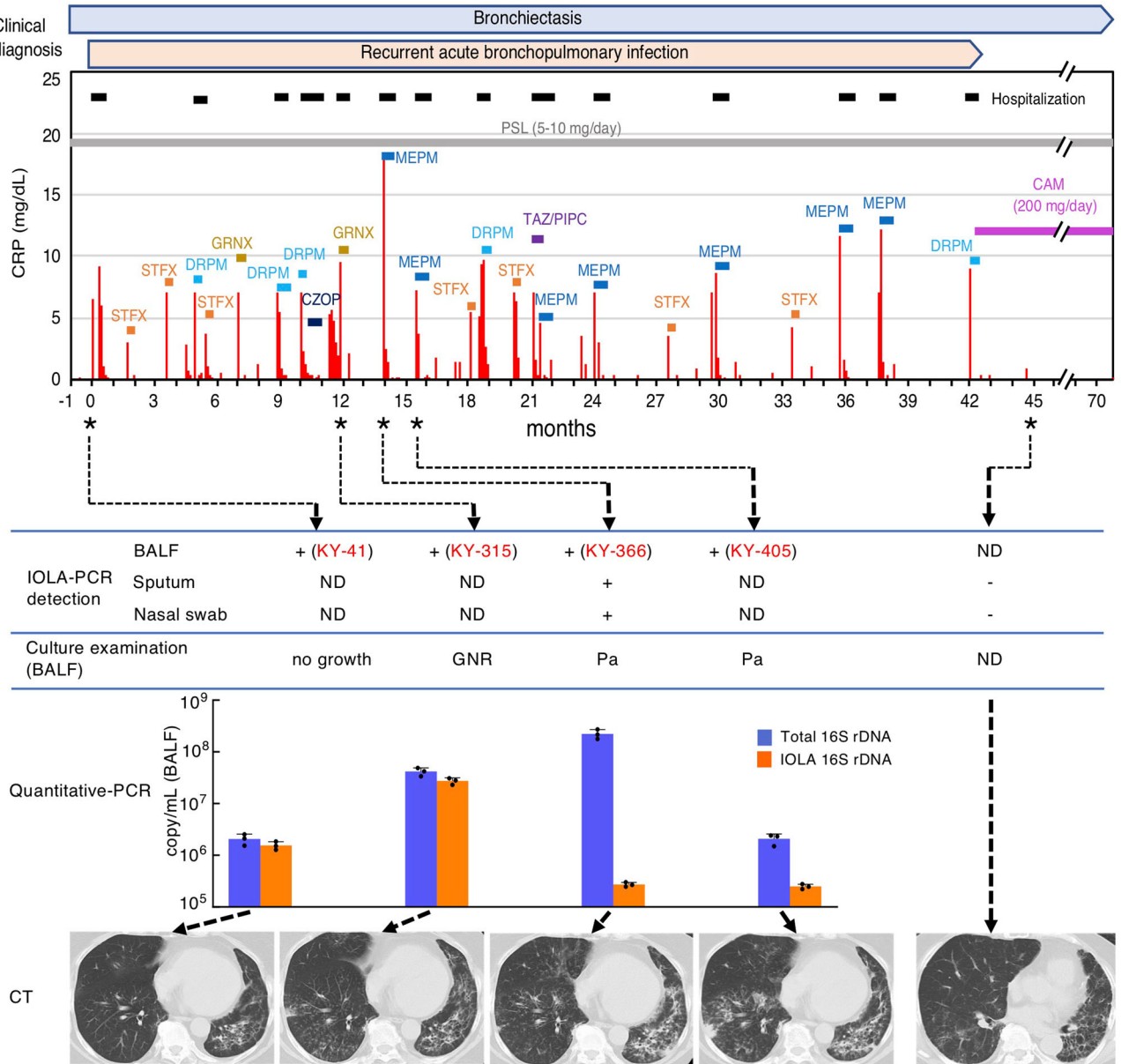

**Fig. 7 Clinical course of a patient with long-term persistence of IOLA.** From the sampling date of KY-41, the clinical history of a patient who yielded four IOLA-positive BALF samples is shown. Black bars at the top represent the periods of hospitalization. Colored bars indicate the durations of antibiotic treatments. Asterisks indicate the dates of BALF specimen sampling. The results of IOLA-specific PCR detection, quantitative PCR analyses, and routine culture examination of BALF, sputum, and nasal swab specimens are presented under the clinical course. Total bacterial rDNA and IOLA 16 S rDNA were quantified by quantitative PCR. CT images of the patient's lower bilateral lobes obtained on each sampling day are also shown. PSL prednisolone (corticosteroid), STFX: sitafloxacin, GRNX garenoxacin, DRPM doripenem, CZOP cefozopran, MEPM meropenem, TAZ/PIPC tazobactam/piperacillin, CAM clarithromycin. + PCR positive, − PCR negative. ND not done, GNR Gram-negative rod, Pa *P. aeruginosa*.

**Long persistence of IOLA in a single patient**. We obtained four IOLA-positive samples (KY-41, KY-315, KY-366, and KY-405) from a single patient who had bronchiectasis and was taking prednisolone (PSL) for treating drug-induced interstitial pneumonia (Fig. 7). This patient recurrently exhibited signs of acute bronchopulmonary infection, such as fever, cough, purulent sputum, and elevated inflammation markers, such as C-reactive protein (CRP), for 42 months until the start of long-term clarithromycin (CAM) treatment as the long-term macrolide antibiotic treatment for bronchiectasis[34] (Supplementary Table 5). In every episode except the first episode, the patient was treated with fluoroquinolones (FQs) or beta-lactams (including carbapenems). BALF samples were obtained four times, i.e., at the first, 9th, 10th,

and 11th episodes, and IOLA was detected in all samples. As described above, no SNPs were detected between these IOLA samples. Quantification of the IOLA 16S rDNA in these BALF specimens revealed that more than $10^5$ copies (per 1 mL of BALF) were present in all specimens. These findings indicate that a single IOLA clone persisted in this patient for at least 15 months. Although FQs and beta-lactams were prescribed several times during this period, they were not able to eliminate IOLA from this patient. Nonsusceptibility of IOLA to beta-lactams is predictable because IOLA lacks the peptidoglycan biosynthesis pathway. Analysis of the GyrA sequence of IOLA revealed that it contains two amino acid substitutions at the residues in the quinolone resistance-determining region (QRDR), whose mutations,

including the Ala67Ser substitution[35], are known to confer quinolone resistance (Supplementary Fig. 10). Thus, it is very likely that IOLA is not susceptible to FQ.

Among the four episodes in which IOLA-positive samples were obtained, *P. aeruginosa* was predominantly detected in the 10th and 11th episodes by culture examination. Accordingly, the copy number ratios of the IOLA 16S rDNA relative to the total bacterial 16S rDNA were 12.0% and 7.5%, respectively. Thus, these episodes were likely caused by *P. aeruginosa* infection. In contrast, no bacteria were cultured in the 1st episode, only a small number of gram-negative rods were detected in the 9th episode, and the copy number ratios of the IOLA 16S rDNA were 75.1% and 66.0%, respectively. While the symptoms suggesting respiratory infection disappeared without prescription of antimicrobials in the first episode and FQ (garenoxacin) was prescribed in the 9th episode, it is unknown whether and how FQ affected the course of this episode.

In the episodes other than the first, 9th, 10th, and 11th episodes, no samples were available for IOLA detection, but routine culture examination of sputum yielded several bacterial pathogens or potential pathogens, such as *P. aeruginosa*, methicillin-resistant *S. aureus* (MRSA), *Neisseria* sp., *Stenotrophomonas maltophilia*, *Moraxella catarrhalis*, and *H. influenzae* (Supplementary Table 5). In these episodes, the patients were treated with FQ or beta-lactams, and the symptoms disappeared quickly. In the examination of the sputum and nasal swab samples obtained 2.6 months after the start of CAM prescription, IOLA was not detected by IOLA-specific PCR (note that IOLA DNA was detected in the sputum and nasal swab samples in the 10th episode in which an IOLA-positive BALF sample was obtained). Thus, IOLA was eliminated from the patient by this time point. However, it is unknown when IOLA was eliminated and whether CAM contributed to the elimination. The computed tomography (CT) image of the bilateral lower lobes was also significantly improved at this time point, and nodular shadows and infiltrations observed in the four episodes were not clearly observed (Fig. 7, at the bottom). Moreover, after starting long-term CAM treatment, the patient experienced no episodes of acute respiratory infection.

## Discussion

By developing a protocol to selectively amplify extremely AT-rich bacterial genomes, we successfully reconstructed the complete IOLA genome sequence and showed that, to the best of our knowledge, IOLA has the smallest genome (304 kb in size) among the human-associated bacterial genomes sequenced to date, followed by *M. genitalium* (580 kb)[36]. The GC content (20.7%) is also extremely low, while slightly higher than those of several insect endosymbionts[31,37–39]. Although only half of the CDSs showed sequence homology to known proteins, even when using a low threshold, fine-scale phylogenetic analyses based on identified housekeeping genes revealed that IOLA is a unique lineage forming a long branch in order Rickettsiales of class Alphaproteobacteria. In Rickettsiales, IOLA is most closely related to family *Anaplasmaceae* but branched off early from this family and likely represents a distinct lineage at the family level.

All bacteria thus far identified as having extremely small genomes (<0.5 Mb) are insect endosymbionts. The genome size of IOLA is comparable to those of these insect endosymbionts. Accordingly, similar to the pattern in insect endosymbionts, numerous metabolic pathways, including those for cell wall synthesis, are missing in IOLA. However, in a clustering analysis based on gene composition, IOLA clustered with members of phylum *Tenericutes*, such as mycoplasma, but not with insect endosymbionts or intracellular parasites in order Rickettsiales and other taxa, suggesting that IOLA has an epicellular parasitic lifestyle similar to that of mycoplasma[32]. In fact, IOLA completely lacks the genes for transport/metabolism of amino acids, in sharp contrast to insect endosymbionts, which possess many genes for amino acid biosynthesis to supply essential amino acids to host[37]. This presumption is supported by our previous failure to detect IOLA cells in host cells via fluorescence in situ hybridization examination[9]. However, to verify this hypothesis, further investigations, including in vitro cultivation and extensive histological examinations of IOLA, are required, although such analyses have not yet been successful.

As for the host dependency of IOLA metabolism, the presence of an ATP/ADP translocase and an $H^+$-PPase is particularly intriguing. Although the bacteria possessing ATP/ADP translocase(s), such as chlamydia, had long been considered "energy parasites" that fully depend on host cell metabolism for energy production, it has recently been reported that while *Chlamydia trachomatis* uses the ATP/ADP translocase at the early stage of infection, it uses its own respiratory metabolism coupled to sodium-dependent ATP synthesis at later stages[40]. It is also possible that IOLA is able to both uptake external ATP and produce ATP using F-type ATP synthase. In this regard, the $H^+$-PPase may be involved in the formation of a proton gradient in IOLA, as the genes for the electron transfer system were not identified.

Another important aspect of IOLA is its association with human respiratory tracts. We identified 14 IOLA-containing lower respiratory tract specimens from 11 patients with respiratory infection, while all patients had some underlying diseases. More importantly, we found that a single IOLA clone persisted in a single patient for at least 15 months. In this patient, acute bronchopulmonary infections occurred repeatedly, and IOLA was detected in all four episodes (the first, 9th, 10th, and 11th episodes), where the presence of IOLA was investigated. The main causative agent of the 10th and 11th episodes was *P. aeruginosa* infection. However, IOLA was predominantly detected and other potential pathogens were not detected in the first and 9th episodes. Although further investigations are necessary to understand the role of IOLA in human respiratory tracts, these observations suggest that IOLA may be associated with specific human respiratory tract conditions, probably in immunocompromised hosts or those with chronic pulmonary diseases. It has been shown that very low numbers of bacteria can be present in the lungs of healthy humans[41,42], but the amounts of IOLA in most IOLA-positive BALF samples from the other 10 patients (8/10) were >$10^4$ copies/mL. Therefore, although many of these acute bronchopulmonary infections identified might not be caused by IOLA, it is strongly suggested that IOLA is not a contaminating agent but an inhabitable bacterium in respiratory tract of patients with respiratory disorders. The ability of IOLA to persist for a long time in the respiratory tract and its intrinsic resistance to beta-lactams and FQs might also be key factors in these cases.

Considering the nature of IOLA as a human-associated bacterium, important issues to be addressed include the mechanisms and factors underlying the host–IOLA interaction. For investing them, it is necessary to establish a cultivation method of IOLA. In addition, the parasitic lifestyle of IOLA inferred in the current study and its potential ability to inhabit animals other than human as inferred in a previous study[11] suggest the presence of a reservoir(s) and a vector(s) for IOLA in the environment. Identification of these organisms or agents is also necessary to

understand the biology and clinical importance of IOLA. In future studies aiming to clarify these issues, the selective conservation of T6SS in IOLA, despite the high level of genome reduction, may be crucial because this system can mediate not only interbacterial interactions but also interactions with eukaryotic hosts[30].

## Methods

**Clinical specimens**. This study was approved by the human and animal ethics review committee of the University of Occupational and Environmental Health (UOEH), Japan (No. 09-118). Written informed consent was obtained from all patients in this study and all relevant ethical regulations were followed. BALF specimens were collected from pulmonary pathological lesions using bronchoscopy and 40 mL of sterile saline[42]. Before BALF specimen sampling, gargling with povidone iodine solution was performed to minimize contamination by oral bacteria. Then, a fiberoptic bronchoscope was introduced orally into the trachea without any contacts to avoid oral bacterial contamination. A total of 523 clinical specimens were obtained from the UOEH-hospital, and its referring hospitals between September 2010 and August 2016. The 523 specimens comprised 479 BALF and 44 sputum samples, which included 252 BALF and 20 sputum samples that we previously analyzed by a clone library-based approach[9].

**Genomic DNA preparation and amplification**. The genomic DNA of IOLA was extracted and amplified from a BALF specimen of patient KY-405 as outlined in Fig. 1. The BALF specimen (400 μL) was first filtered through a polytetrafluoroethylene (PTFE) filter with a 1.2-μm pore size (Sartorius Stedim) to remove host cells and larger bacterial cells. The filtrate was then passed through a filter with a 0.2-μm pore size (Advantec). After being washed twice with 1000 μL of PBS, the filter was transferred to a 15-mL tube and incubated with 2 mL of PBS containing 2% SDS at room temperature for 10 min. After phenol extraction (two times), the DNA in the aqueous phase was concentrated with an Amicon Ultra 0.5 mL 100 K filter (Merck Millipore) and amplified with an Illustra Genomiphi DNA Amplification Kit (GE Healthcare Life Sciences) according to the manufacturer's instructions except in relation to the heat denaturation process. To selectively amplify the extremely AT-rich genomic DNA of IOLA, we optimized the denaturation conditions to 80 °C for 3 min. The amplified DNA was purified using the Wizard DNA Clean-Up System (Promega).

**Genome sequence analyses**. The genome sequence of IOLA was determined by a combined strategy of Illumina and Sanger sequencing[43] (Supplementary Fig. 1). Sequencing libraries were prepared following the manufacturer's protocol for the Nextera XT Library Preparation Kit and the Nextera Mate Pair Sample Preparation Kit (Illumina). Sequencing with an Illumina MiSeq sequencer generated paired-end reads (250 bp x2) and mate-pair reads (2–4-kb insert length). After filtering low-quality reads and human-derived reads using FastQC (Illumina) and BWA[44], the remaining sequence reads were assembled using SPAdes[12] and Platanus[13], respectively. Scaffolds with a low GC content (<25%) and lengths of >1000 bp generated by the two assemblers were aligned and merged into nine scaffolds using ATSQ5.1.3 (GENETYX Corporation). Gap closing and resequencing of low-quality regions were achieved by PCR amplification and Sanger sequencing (Supplementary Fig. 1). In the examination of scaffolds which were not used in the final assembly (Supplementary Fig. 2), blastn searches were performed against the latest human genome database (GCF_000001405.39_GRCh38.p13 dataset) and the NCBI nt database (updated September 29, 2020).

The genome sequence was first annotated with the Microbial Genome Annotation Pipeline (MiGAP) (http://www.migap.org/), implementing MetaGeneAnnotator ver. 1.0[45] and tRNAscan-SE ver. 1.23[46] to identify CDSs and tRNA genes, respectively, followed by manual inspection and correction of all start codons. The functions of each CDS were predicted based on the results of a homology search using NCBI blastp[47], and the reference proteins dataset (refseq_protein) with the confidence threshold values set to a >50% overlap length and >25% identity. Domain and motif searches were performed for each CDS using Pfam[29] and TIGRfam[48] on the HMMER website (https://www.ebi.ac.uk/Tools/hmmer/). When a CDS was found to contain a conserved domain in a known protein family (cutoff E-value; 2.0E−7), the protein family name was assigned to the CDS, even if the results of the blastp search did not satisfy the threshold value. Secondary structures and subcellular localization of the CDSs were predicted by the SOSUI[23] and PSORTb[24] programs. The rRNA gene sequences were analyzed using Rfam[49] on the EMBL-EBI website (https://www.ebi.ac.uk/services/dna-rna). For genomic comparison with representative bacteria with extremely small genomes, complete genomes smaller than 600 kb were downloaded from the NCBI website (https://www.ncbi.nlm.nih.gov). When multiple genomes were sequenced in the same species, the first one sequenced was selected. The COG datasets of these bacterial genomes were obtained from the IMG/M database[50] of the Joint Genome Institute (https://img.jgi.doe.gov). COG assignment of IOLA CDSs was conducted using CD-search[51] with the COG v1.0-4873PSSMs database (threshold: E-value <0.01). The 17 CDSs (IOLA_001, 008, 018, 042, 051, 080, 105, 128, 145, 163, 168, 171, 186, 198, 244, 271, and 278) that were not assigned by the CD search were categorized in accordance with their annotations.

**Phylogenetic analyses**. Classification of IOLA based on the 16S rRNA gene sequence was attempted using the RDP Classifier[14] with a confidence threshold of 60%. The similarity search of the 16S rRNA gene was performed using NCBI blast[47] (discontiguous megablast algorithm) with four NCBI databases (https://www.ncbi.nlm.nih.gov): the nucleotide collection (nr/nt), the 16S rRNA sequence collection (Bacteria and Archaea), the RefSeq prokaryote representative genome collection, and the High Throughput Genomic Sequence (HTGS) collection.

For other phylogenetic analyses, the datasets of amino acid and nucleotide sequences were collected from the IMG/M database[50] (Supplementary Data 2) or the NCBI database, multiple alignments were built using MAFFT[52] with the default settings, and poorly aligned and divergent regions were eliminated using Gblocks 0.91b[19]. IQ-tree[53] was used to construct maximum likelihood (ML) trees, with suitable models selected using the "Model finder" tool in IQ-tree. Trees were visualized with iTOL (v 5.5.1)[54].

*Tree of life based on ribosomal protein sequences*. The tree was constructed based on the concatenated sequences of 16 ribosomal proteins (RpL2, 3, 4, 5, 6, 14, 15, 16, 18, 22, and 24 and RpS3, 8, 10, 17, and 19) as described previously[3]. For tree construction, we selected 122 bacterial genomes from the IMG/M database[50] (Supplementary Data 2). They represented 56 bacterial phyla, and each genome contained at least 14 of the 16 ribosomal proteins. Using the resulting 1895-position alignment, an ML tree was constructed using the LG+ R10 model and 1000 ultrafast bootstrap replicates[55].

*Tree of life based on rRNA gene sequences*. Among the 122 abovementioned bacteria, 92 for which nearly full-length 16S and 23S rRNA gene sequences (>1200 bp and >2000 bp, respectively) were available[50] were selected, and phylogenetic analysis was performed based on their concatenated 16S and 23S rRNA sequences. Using the 3421-position alignment, an ML tree was constructed with the TIMe +R10 model and 1000 ultrafast bootstrap replicates[55].

*Analysis of proteobacteria*. An ML tree was constructed using 451 species of phylum Proteobacteria, 2 species of phylum Actinobacteria (outgroup), and IOLA (Supplementary Data 2) based on their concatenated sequences of the 16 ribosomal proteins (2016-position alignment) using the LG + R10 model and 1000 ultrafast bootstrap replicates.

*Analysis of alphaproteobacteria*. A tree was constructed using 165 species, including 157 species of Alphaproteobacteria, seven species of Zetaproteobacteria (outgroup), and IOLA (Supplementary Data 2), based on their concatenated sequences of the 16 ribosomal proteins (1872-position alignment) using the LG + F + R10 model and 1000 ultrafast bootstrap replicates. A tree based on the concatenated sequences of 12 housekeeping proteins (SecA, SecY, TsaD, RpoA, RecA, DnaK, GroEL, GyrA, GyrB, PheS, SerS, and LeuS) was also constructed using the 5048-position alignment, as described above.

*Analysis of alphaproteobacteria using BMGE and PMSF*. The MAFFT alignment was trimmed with BMGE version 1.12 using the BLOSUM62 similarity matrix with a block size of 5. The resulting 2121-position alignment of 16 ribosomal proteins and the 5624-position alignment of 12 housekeeping proteins were applied for ML tree construction using the PMSF model (LG + C20 + F + R10) with 100 bootstrap replicates.

*Analysis of PPases*. A Bayesian tree was generated using MrBayes 3.2.6[56] on the CIPRES Science Gateway (https://www.phylo.org/). The 121 PPase protein sequences used in a previous paper[25] were collected from the NCBI database (Supplementary Table 6) and aligned with the IOLA PPase using default settings in MAFFT, and ambiguously aligned residues were removed using Gblocks 0.91b. Bayesian analysis using the "mixed" amino acid model was conducted with two independent runs, each with four chains that were 10,000,000 generations long. Trees were sampled every 1000 generations, and the first 25% of trees were discarded as burn-in. The convergence of runs was evaluated by the average standard deviation of split frequencies (less than 0.01).

*Analysis of ATP/ADP translocases*. A total of 54 ATP/ADP translocase sequences, including 37 sequences identified by blastp search of the UniProtKB/SwissProt database using the ATP/ADP translocase of IOLA as a query, 11 sequences from phylum TM6, five sequences of *Ca*. Liberibacteria, and that of IOLA, were used (Supplementary Fig. 7). The MAFFT alignment was trimmed by Gblocks 0.91b, and the resulting alignment (212 residue columns) was applied for ML tree construction using IQ-tree (the mtZOA+G4 model with 1000 ultrafast bootstrap replicates).

**Genome comparison of IOLA and other bacteria with small genomes and low GC contents**. To evaluate the genome size and GC content of IOLA, 9870

complete bacterial genomes were collected from the IMG/M database. All information, such as the number of genes in each COG functional category, host names, genome sizes, and ecosystems, was also collected from the IMG/M database. From this dataset, 265 host-associated bacteria with genomes smaller than 1.2 Mb were selected, and 39 bacterial strains with the smallest genome in each genus were further selected. Using the COG information of these 39 genomes (Supplementary Data 3 and 4), a clustering analysis based on the proportion of COG functional categories was performed with R software ver. 3.5.0 (http://cran.r-project.org).

**PCR detection of the IOLA 16S rRNA gene.** The presence of IOLA in clinical specimens was assessed by IOLA-specific nested PCR designed in this study. DNA used as templates was extracted from specimens as follows. An aliquot of BALF (400 μL) was mixed with 500 μL of TE buffer, 100 μL of 30% SDS solution (final concentration 3%), and 0.3 g of glass beads. The mixture was then vigorously shaken using a Micro Smash MS-100 apparatus (Tomy Seiko Co., Ltd.). After phenol–chloroform–isoamyl alcohol (25:24:1, vol/vol) extraction (two times), DNA in the aqueous phase was washed twice with PBS buffer and TE buffer, respectively, and concentrated to 30 μL using an Amicon Ultra 0.5 mL 100 K filter (Merck Millipore).

The reaction mixtures (10 μL in total volume) for the 1st PCR contained each of the bacterial universal primers (E341f and E907r) at 100 nM, 1 μL of extracted DNA and 5 μL of AmpliTaq Gold DNA 360 Polymerase Master Mix (Applied Biosystems), and amplification was achieved by denaturing at 96 °C for 5 min, followed by 25 cycles of 96 °C for 30 s, 53 °C for 30 s, and 72 °C for 1 min and a final elongation step at 72 °C for 2 min. GeneAmp 9700 (Applied Biosystems) was used for amplification. After the 1st PCR, the reaction mixture was diluted 10-fold with TE buffer and used as a template for the 2nd PCR, which was performed using the IOLA 16S rRNA gene-specific primers (IOLA-F1 and IOLA-R0)[9]. The reaction mixtures for the 2nd PCR (10 μL in total volume) contained the IOLA-specific primers (100 nM each), 1 μL of the template, and 5 μL of AmpliTaq Gold DNA 360 Polymerase Master Mix. Amplification was performed as described for the 1st PCR except that 30 amplification cycles were employed. PCR amplicons were examined by 2% agarose gel electrophoresis. For detection limit analysis, a nearly full-length IOLA 16S rRNA gene was amplified from the BALF specimen of KY-405 using a specific primer pair (IOLA-27F: AGGAGTTTGATCCTGGCTCAG and IOLA-nR1: GTCAAAAGCGCAGGTTCAC) and AmpliTaq Gold DNA 360 Polymerase Master Mix. Amplification was achieved by denaturing at 96 °C for 5 min, followed by 30 cycles of 96 °C for 30 s, 55 °C for 30 s, and 72 °C for 2 min and a final elongation step at 72 °C for 4 min. To prepare a positive control, the PCR-amplified IOLA 16S rRNA gene was inserted into the pCR4-TOPO plasmid (Invitrogen). The plasmid DNA was purified from *Escherichia coli* Top10 cells transformed with this plasmid, and the DNA concentration was determined by the absorbance at 260 nm using a NanoDrop 2000c instrument (Thermo Fisher Scientific). Finally, serial 10-fold dilutions of the plasmid DNA were prepared and subjected to nested PCR as described above.

**Quantitative PCR analysis of 16S rRNA genes.** The copy numbers of the 16S rRNA gene in the DNA extracted from IOLA-positive specimens were evaluated by using the real-time PCR method. The bacterial universal primer set (E341f and E907r) and the IOLA 16S rRNA gene-specific primer set (IOLA-F1 and IOLA-R0) were used to count total bacterial 16S rRNA genes and the IOLA 16S rRNA gene, respectively. Twenty-microliter reaction mixtures, each containing 1 μL of the extracted DNA solution, each of the primer sets at 100 nmol/L, and 10 μL of Thunderbird SYBR qPCR Mix (Toyobo), were incubated at 95 °C for 1 min, followed by 40 cycles of 95 °C for 15 s, 56 °C for 15 s, and 72 °C for 1 min, in the ABI PRISM 7000 sequence detection system (Applied Biosystems). Standard curves were created using serial 10-fold dilutions of the pCR4-TOPO plasmid containing a nearly full-length IOLA 16S rRNA gene described above. For each total bacterial and IOLA 16S rDNA analysis, tests were performed in triplicate (in three independent experiments).

**PCR amplification and sequencing analysis of genomic segments of IOLA.** From 14 clinical specimens detected as positive with IOLA-specific nested PCR, 12 genomic segments of IOLA were amplified by nested PCR. The primers for the 1st PCR were designed to amplify an approximately 1.2-kb segment located in the IOLA genome with approximately 25-kb intervals. The 2nd PCR primers were designed approximately 100 bp inside the 1st PCR primers (Supplementary Data 5). The reaction mixture (20 μL final volume) contained 4.0 μL of dNTP mix (containing each dNTP at 2 mM), 0.4 μL of primer mix (2 pmol for each primer), 1 μL of template DNA, 0.4 μl (0.4 U) of KOD FX Neo DNA Polymerase (Toyobo), 10 μL of 2× PCR buffer for KOD FX Neo, and 4.2 μL of ultrapure water. PCR amplification was performed on a GeneAmp 9700 instrument with a program of 94 °C for 2 min, 35 cycles of 94 °C for 10 s, 52 °C for 20 s, and 68 °C for 30 s, and a final elongation step at 68 °C for 1 min. The reaction mixture was diluted 20-fold with TE buffer and used as template for the 2nd PCR. The 2nd PCR was performed in a reaction mixture (20 μL) containing 0.4 μL of inside primers (2 pmol for each primer), 1 μL of the 20-fold-diluted 1st PCR mixture, 10 μL of AmpliTaq Gold DNA 360 Polymerase Master Mix, and 8.6 μL of ultrapure water. PCR amplification was carried out on a GeneAmp 9700 instrument with a program of 96 °C for 5

min, 35 cycles of 96 °C for 30 s, 53 °C for 30 s, and 60 °C for 90 s, and a final elongation step at 60 °C for 3 min. Amplicons were examined by 1.5% agarose gel electrophoresis. Each amplicon obtained by nested PCR was sequenced by the Sanger method using the four sequencing primers listed in Supplementary Data 5. An ML tree was constructed based on the concatenated sequences of the 12 segments obtained from each specimen using MEGA 7.0.26 software for Mac[57], the Tamura 3-parameter model with a gamma distribution and 1000 bootstrap replicates.

**Statistics and reproducibility.** All the results of phylogenetic analyses can be reproduced with the parameters indicated. Quantitative PCR analyses for each total bacterial and IOLA 16S rDNA in specimen were performed three times independently for each sample to calculate the mean and standard deviation.

**Reporting summary.** Further information on research design is available in the Nature Research Reporting Summary linked to this article.

## Data availability
The sequence read data of IOLA KY-405 have been deposited in DDBJ Sequence Read Archive (DRA) under the BioProject identifier PRJDB7462. The assembled IOLA genome sequence has been deposited in DDBJ/EMBL/GenBank under accession No. AP019313. The 16S rRNA gene sequences (accession No. LC435447-LC435456) and PCR amplicon sequences (accession No. LC438941-LC439108) from the IOLA-positive specimens have also been deposited in DDBJ/EMBL/GenBank.

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

## Acknowledgements

This work was supported by KAKENHI (15K15136 and 20K07491 to K.F.) from the Japan Society for the Promotion of Science (JSPS).

## Author contributions

K.F. and T.H. designed the study and wrote the manuscript. K.Yam., T.K., S.N., K.Yat., H.I., K.A., and H.M. collected clinical specimens and evaluated clinical significance of the results. K.F. and H.T. conceived the experiments regarding DNA amplification and quantification of IOLA. K.F., Y.O., K.K., and T.H. performed DNA sequencing and analyzed DNA sequence data.

## Competing interests

The authors declare no competing interests.
