## [Peer Review File · Communications Biology]

Reviewers' comments:

Reviewer #1 (Remarks to the Author):

In this submitted manuscript by Fukuda and colleagues, the authors examined the presence of a novel bacterium in specimens from patients with a range of respiratory conditions and co-morbidities including bronchiectasis, allergic bronchopulmonary aspergillosis, bronchial asthma, interstitial pneumonia, diabetes mellitus, rheumatoid arthritis, and hypertension. While the manuscript investigates an interesting topic, there are areas that require careful attention regarding the submission in its current form.

Main Comments

1. In regard to the novel bacterium, the authors state on Line 48 that they "temporarily named it Infectious Organism Lurking in human Airways (IOLA)". Without establishing the association between the bacterium and human disease, it appears premature to refer to it as an "infectious" organism. Its presence in the respiratory system of 11 individuals in this study would not amount to causation of an infectious disease. Further data would be required to validate the role of the bacterium in specific human disease(s) and its status as a transmissible infectious agent.

2. In the genome sequence analysis of the "IOLA" bacterium from sample KY-405, the authors state that they aligned and merged the sequences into 9 nine scaffolds from which gaps and ambiguous regions were removed. What is not clear from the genome assembly description is how the authors ensured that their genome assembly was not missing any sequence of the novel bacterium.

The authors note that "Based on COG assignment, no genes categorized into "RNA processing and modification", "carbohydrate transport and metabolism", "coenzyme transport and metabolism", "cell motility", "cytoskeleton", "mobilome" or "secondary metabolite biosynthesis" were identified in IOLA." In addition, they state that "Despite the apparent lack of the ability to synthesize amino acids, carbohydrates, phospholipids, and coenzymes, only a few genes involved in the transport of these compounds were identified, suggesting the possibility that IOLA contains previously unknown systems for the uptake of these compounds."

It is essential to be assured that the authors have not missed sequence in the generation of their genome for the sample KY-405.

In addition, although the authors enriched their metagenomic DNA for AT-rich DNA, it is not made clear in the manuscript how the authors ensured that no sequences of foreign DNA from their metagenome preparation ended up in their final genome assembly sequence.

3. It appears from the manuscript that comparisons of different samples of the "IOLA" bacterium were made using 12 amplified genomic segments, approximately 1 kb in size each, providing 12,410 bp sequence out of 303,838 bp. Given the substantial claim being made in the manuscript i.e. the discovery of the "the smallest and most AT-rich genome among known human-associated bacterial genomes", it is perhaps surprising that the authors have not described in the manuscript the application of their protocol for DNA isolation, enrichment and whole genome sequencing to multiple samples of the bacterium, in particular, the 4 longitudinal samples from the single patient with bronchiectasis. An analysis of the whole genomes assembled from these longitudinal samples would assist in validating aspects of the genome sequence including the completeness of the sequence coverage and % AT content.

4. In the specimen collection, the authors state that they "screened a total of 523 specimens, including 480 BALF and 43 endotracheal aspirate (EA) specimens, from 501 patients with respiratory diseases, such as pneumonia, bronchitis, and lung abscess." It is not clear from the manuscript how much screening was performed of specimens from the respiratory tracts of healthy individuals. The authors appear to be associating the presence of the "IOLA" bacterium

with respiratory disease e.g. their statement on Line 359: "Another important aspect of IOLA is its association with human respiratory infection." But there are insufficient data provided in the manuscript to show that the "IOLA" bacterium is present at a significantly higher frequency in the respiratory tracts of patients with respiratory disease compared to healthy individuals.

Additional Comments

Line 220. "linage" should be "lineage".

Line 293. "GryA" should be "GyrA".

Reviewer #2 (Remarks to the Author):

Fukuda et al. report the full genome sequence and genomic characterization of a newly-identified bacterium, IOLA, that was recovered from human bronchoalveolar fluid. The unusual genomic features of IOLA – its small size, high AT content, and very low homology to known bacteria – make this a very intriguing study, and also set a high threshold to ensure its validity.

1. The manuscript would benefit from additional description of methods, in particular: were other strategies (amplification-free sequencing) attempted to manage high AT content? Also, it seems to me that long-read sequencing would be extremely beneficial in confirming this unusual genome; I cannot tell to what extent this may have been achieved by the use of mate pair reads, but what about PacBio or alternative strategies for long read sequencing?
2. There is really no evidence to suggest that this organism is a pathogen, since the human lung is not a sterile environment especially in patients with bronchiectasis. I recommend modulating the language around this throughout the paper, and in particular I do not think it is appropriate to state that IOLA was mainly involved (line 305) or had a major contribution (367) to clinical exacerbations.
3. I think it is likely that the long branch lengths in phylogenetic analysis do not suggest extremely rapid genetic evolution (lines 131 and 146) but instead reflect the fact that the genome is so divergent from the included references. Are there more closely-related references that can be included (eg the ones with ~73% homology, reported on line 111).

Responses to the comments from Reviewer 1:

Thank you for your valuable and thoughtful comments. We carefully revised our manuscript according to your suggestions as follows.

Main comments:

1. *In regard to the novel bacterium, the authors state on Line 48 that they “temporarily named it Infectious Organism Lurking in human Airways (IOLA)”. Without establishing the association between the bacterium and human disease, it appears premature to refer to it as an “infectious” organism. Its presence in the respiratory system of 11 individuals in this study would not amount to causation of an infectious disease. Further data would be required to validate the role of the bacterium in specific human disease(s) and its status as a transmissible infectious agent.*

<Response>

We agree that our current data are insufficient to show the infectivity and/or etiological roles of IOLA. We have changed all terms and phrases suggesting that IOLA is a pathogen (and related terms/phrases) to "a human respiratory tract-associated bacterium" or similar terms/phrases throughout the manuscript. However, regarding the term “IOLA”, we used it in our previous report (ref.9). So, for consistency, we want to keep using it as a tentative name of this bacterium in this paper. Please see the revised manuscript, in which all changes are highlighted in bold red.

Original manuscript Line 25: “a human respiratory tract-associated uncultured bacterium” was changed to “**an uncultured bacterium detected in human respiratory tract**” (Line 25 in the revised manuscript).

Original manuscript Line 33: “suggesting that IOLA is a previously unknown potential human pathogen.” was changed to “suggesting that IOLA is a **human respiratory tract-associated bacterium.**” (Line 33 in the revised manuscript).

Original manuscript Line 63: “human pathogenic bacterial” was changed to “**human-associated bacterial**” (Line 64 in the revised manuscript).

Original manuscript Lines 67-68: “suggest that IOLA may be a previously unknown potential human

pathogen.” was changed to “suggest that IOLA **can inhabit and persist in the respiratory tract of patients with respiratory disorders.**” (Lines 68-69 in the revised manuscript).

Original manuscript Lines 304-305: “These findings suggest the possibility that IOLA was mainly involved in these two episodes. was deleted in the revised manuscript.

Original manuscript Lines 367-369: “9th episodes, suggesting the major contribution of IOLA to the onset of these episodes. These observations suggest the possibility that IOLA has the ability to cause human respiratory diseases,” was changed to “9th episodes. **Although further investigations are necessary to understand the role of IOLA in human respiratory tracts, these observations suggest that IOLA may be associated with specific human respiratory tract conditions,**” (Lines 382-384 in the revised manuscript).

Original manuscript Line 373: “IOLA-associated” was changed to “**these**” (Line 388 in the revised manuscript).

Original manuscript Lines 374-376: “such as the 10th and 11th episodes mentioned above, not only repeated prescription of antibiotics but also the presence of IOLA might enhance or exacerbate infections by other pathogens in these cases.” was changed to “**it is strongly suggested that IOLA is not a contaminating agent but an inhabitable bacterium in respiratory tract of patients with respiratory disorders.**” (Lines 389-391 in the revised manuscript).

Original manuscript Lines 378-382: “Considering the nature of IOLA as a potential human pathogen, important issues to be addressed include the mechanisms and factors underlying the host-IOLA interaction. In addition, the parasitic lifestyle of IOLA inferred in the current study and the possibility of it serving as an agent of zoonotic infection as inferred in a previous study¹¹ suggest the presence of a reservoir(s) for IOLA in the environment and a vector(s) to transmit it to humans.”

was changed to

“Considering the nature of IOLA as a **human-associated bacterium**, important issues to be addressed include the mechanisms and factors underlying the host-IOLA interaction. **For investing them, it is necessary to establish a cultivation method of IOLA.** In addition, the parasitic lifestyle of IOLA inferred in the current study and **its potential ability to inhabit animals other than human**

as inferred in a previous study¹¹ suggest the presence of a reservoir(s) and a vector(s) for IOLA in the environment.” (Lines 393-398 in the revised manuscript).

2. In the genome sequence analysis of the “IOLA” bacterium from sample KY-405, The authors note that It is essential to be assured that the authors have not missed sequence in the generation of their genome for the sample KY-405. the authors state that they aligned and merged the sequences into 9 nine scaffolds from which gaps and ambiguous regions were removed. What is not clear from the genome assembly description is how the authors ensured that their genome assembly was not missing any sequence of the novel bacterium. The authors note that “Based on COG assignment, no genes categorized into ”RNA processing and modification“, ”carbohydrate transport and metabolism“, ”coenzyme transport and metabolism“, ”cell motility“, ”cytoskeleton“, ”mobilome“, ”secondary metabolite biosynthesis“, were identified in IOLA”. In addition, they state that “Despite the apparent lack of the ability to synthesize amino acids, carbohydrates, phospholipids, and coenzymes, only a few genes involved in the transport of these compounds were identified, suggesting the possibility that IOLA contains previously unknown systems for the uptake of these compounds”. It is essential to be assured that the authors have not missed sequence in the generation of their genome for the sample KY-405. In addition, although the authors enriched their metagenomic DNA for AT-rich DNA, it is not made clear in the manuscript how the authors ensured that no sequences of foreign DNA from their metagenome preparation ended up in their final genome assembly sequence.

<Response>

As pointed out by the reviewer, our descriptions on genome assembly process lacked some details. We have provided a new supplementary figure (Fig. S1) showing the process and data related to finishing step (lines 85-92). Please note that the number of SPAdes scaffolds used in the final assembly was incorrect (not 16 but 15). This was corrected in the revised text (lines 88).

In addition, to ensure that our final assembly is not missing any sequences, we performed a systematic analysis of all SPAdes and Platanus scaffolds which we did not use for the final IOLA genome assembly (other than the 40 scaffolds used). As described in the revised text (lines 93-103) and detailed in a new supplementary figure (Fig. S2), we identified (i) the scaffolds showing high similarity to human sequences (threshold; E-value <1.00E-100), (ii) scaffolds having much higher GC content (>50%) than that of IOLA (20.7% on average), and (iii) the scaffolds showing high

similarity to the sequences of other known organisms (E-value <1.00E-100), and these scaffolds were excluded from the data set because they are very unlikely related to IOLA. Among the remaining scaffolds, we identified several scaffolds corresponding to the gap regions of IOLA assembly. They were smaller than 1 kb (n=6) or had a GC content slightly higher than 25% (n=1). Finally, based on the GC content and sequence coverage (sequencing depth) of each scaffold, we confirmed that other remaining scaffolds can be clearly distinguished from the scaffolds used in the final assembly. This result, along with the results of systematic PCR/sequencing analyses of all gaps and ambiguous regions in scaffolds (regions with potential risk of misassembly) shown in Fig. S1, indicates that it is very unlikely that our final assembly are missing some sequences and contains some unrelated sequences.

Lines 85-92 in the revised manuscript: “We selected scaffolds of >1,000 bp in length and with a low GC content (<25%). From this scaffold sets, the scaffolds which were found to be derived from known organisms or to be low coverage (less than 5x) were removed, yielding 15 and 25 scaffolds from SPAdes and Platanus assemblies, respectively (Supplementary Fig. 1). The 40 scaffolds were aligned and merged into nine scaffolds. By closing the gaps (n=9) and ambiguous regions (n=18) in these scaffolds by PCR and Sanger sequencing (see Supplementary Fig. 1 for details), we obtained a circular 303,838-bp chromosome of IOLA KY-405 with an average GC content of 20.7% (Fig. 2a).”

Lines 93-103 in the revised manuscript: “To verify that the final assembly is not missing any sequence of IOLA, we analyzed, all SPAdes and Platanus scaffolds that were not used for the final assembling step (see Supplementary Fig. 2 for details) and found that most of them (89.2% of the SPAdes scaffolds and 92.7% of the Platanas scaffolds) showed high sequence similarities to the sequences of human or other known organisms (E-value; <1.00E-100) and/or had high GC contents (>50%). Among the remaining scaffolds, several scaffolds were found to correspond to the gap regions of IOLA assembly. They were smaller than 1 kb (n=6) or had a GC content slightly higher than 25% (n=1). Other scaffolds were clearly distinguished from the scaffolds used in the final assembly based on GC content and sequencing coverage (depth). Thus, it is very unlikely that the final assembly is missing some sequences of IOLA.”

Lines 441-445 in the revised manuscript: “(Supplementary Fig.1). In the examination of scaffolds

which were not used in the final assembly (Supplementary Fig.2), blastn searches were performed against the latest human genome database (GCF_000001405.39_GRCh38.p13 dataset) and the NCBI nt database (updated September 29, 2020).”

3. It appears from the manuscript that comparisons of different samples of the “IOLA” bacterium were made using 12 amplified genomic segments, approximately 1 kb in size each, providing 12,410 bp sequence out of 303,838 bp. Given the substantial claim being made in the manuscript i.e. the discovery of the “the smallest and most AT-rich genome among known human-associated bacterial genomes”, it is perhaps surprising that the authors have not described in the manuscript the application of their protocol for DNA isolation, enrichment and whole genome sequencing to multiple samples of the bacterium, in particular, the 4 longitudinal samples from the single patient with bronchiectasis. An analysis of the whole genomes assembled from these longitudinal samples would assist in validating aspects of the genome sequence including the completeness of the sequence coverage and % AT content.

<Response>

To be honest, we also wanted to sequence other BALF specimens. Unfortunately, however, we were unable to apply the protocol shown in Fig. 1 to other specimen due to the lower amounts and qualities (high viscosity and large amounts of DNA from host and other bacteria). We briefly explained this point in the revised manuscript (lines 273-276).

“---because we were unable to apply the protocol shown in Fig. 1 to other specimen due to the lower amounts and qualities (high viscosity and large amounts of DNA from host and other bacteria).”

And we added the sentence regarding to DNA extraction.

“The DNAs extracted from specimens as described^{9,42} were used as templates for the each of the PCR reactions.” (Line540-541 in the revised manuscript)

4. In the specimen collection, the authors state that they “screened a total of 523 specimens, including 480 BALF and 43 endotracheal aspirate (EA) specimens, from 501 patients with respiratory diseases, such as pneumonia, bronchitis, and lung abscess.” It is not clear from the manuscript how much screening was performed of specimens from the respiratory tracts of healthy individuals. The authors appear to be associating the presence of the “IOLA” bacterium

with respiratory disease e.g. their statement on Line 359: “Another important aspect of IOLA is its association with human respiratory infection.” But there are insufficient data provided in the manuscript to show that the “IOLA” bacterium is present at a significantly higher frequency in the respiratory tracts of patients with respiratory disease compared to healthy individuals.

<Response>

We analyzed no BALF samples from healthy individuals as it is difficult to conduct BALF examination on healthy individuals. So, we can not show that IOLA is present at a significantly higher frequency in the respiratory tracts of patients with respiratory disease compared to healthy individuals. We changed this sentence to " Another important aspect of IOLA is its association with human respiratory **tracts**." (Line 374 in the revised manuscript). Other related terms/phrases were also changed as we described in our response to the reviewer's first comment.

Additional Comments:

Line 220. “Linege”should be “Lineage”, Line 293. “GryA” should be “GyrA”

<Response>

Thank you for pointing it out. We have corrected both typos. (Line 234 “**lineage**” and Line 309 “**GyrA**” in the revised manuscript)

Responses to the comments from Reviewer 2:

Thank you for your valuable and thoughtful comments. We carefully revised our manuscript according to your suggestions as follows.

1. The manuscript would benefit from additional description of methods, in particular: were other strategies (amplification-free sequencing) attempted to manage high AT content? Also, it seems to me that long-read sequencing would be extremely beneficial in confirming this unusual genome; I cannot tell to what extent this may have been achieved by the use of mate pair reads, but what about PacBio or alternative strategies for long read sequencing?

<Response>

We did not employ other strategies (e.g., amplification-free shotgun sequencing) because, if selective enrichment by amplification was not used, the ratio of IOLA DNA in the sample was too low. Long-read sequencing using PacBio or Nanopore sequencers is much more powerful than mate pair sequencing in general, but it requires a large amount of DNA. So, it is difficult to apply these technologies to our samples (at least at present). However, we are planning to apply amplification-free shotgun sequencing to the IOLA-positive specimens which we are now trying to collect, because we can now produce much larger amounts of reads at a lower cost by using NovaSeq and because the read mapping strategy to the reference can be effective.

2. There is really no evidence to suggest that this organism is a pathogen, since the human lung is not a sterile environment especially in patients with bronchiectasis. I recommend modulating the language around this throughout the paper, and in particular I do not think it is appropriate to state that IOLA was mainly involved (line 305) or had a major contribution (367) to clinical exacerbations.

<Response>

We agree that our data are insufficient to suggest that IOLA is a pathogen. We have changed all terms and phrases suggesting that IOLA is a pathogen (and related terms/phrases) to "a human respiratory tract-associated bacterium" (or similar terms/phrases) throughout the manuscript.

Please see the revised manuscript, in which all changes are highlighted in bold red.

Original manuscript Line 25: "a human respiratory tract-associated uncultured bacterium" was

changed to “**an uncultured bacterium detected in human respiratory tract**” (Line 25 in the revised manuscript).

Original manuscript Line 33: “suggesting that IOLA is a previously unknown potential human pathogen.” was changed to “suggesting that IOLA is a **human respiratory tract-associated bacterium**.” (Line 33 in the revised manuscript).

Original manuscript Line 63: “human pathogenic bacterial” was changed to “**human-associated bacterial**” (Line 64 in the revised manuscript).

Original manuscript Lines 67-68: “suggest that IOLA may be a previously unknown potential human pathogen.” was changed to “suggest that IOLA **can inhabit and persist in the respiratory tract of patients with respiratory disorders**.” (Lines 68-69 in the revised manuscript).

Original manuscript Lines 304-305: “These findings suggest the possibility that IOLA was mainly involved in these two episodes. was deleted in the revised manuscript.

Original manuscript Lines 367-369: “9th episodes, suggesting the major contribution of IOLA to the onset of these episodes. These observations suggest the possibility that IOLA has the ability to cause human respiratory diseases,” was changed to “9th episodes. **Although further investigations are necessary to understand the role of IOLA in human respiratory tracts, these observations suggest that IOLA may be associated with specific human respiratory tract conditions**,” (Lines 382-384 in the revised manuscript).

Original manuscript Line 373: “IOLA-associated” was changed to “**these**” (Line 388 in the revised manuscript).

Original manuscript Lines 374-376: “such as the 10th and 11th episodes mentioned above, not only repeated prescription of antibiotics but also the presence of IOLA might enhance or exacerbate infections by other pathogens in these cases.” was changed to “**it is strongly suggested that IOLA is not a contaminating agent but an inhabitable bacterium in respiratory tract of patients with respiratory disorders**.” (Lines 389-391 in the revised manuscript).

Original manuscript Lines 378-382: “Considering the nature of IOLA as a potential human pathogen, important issues to be addressed include the mechanisms and factors underlying the host-IOLA interaction. In addition, the parasitic lifestyle of IOLA inferred in the current study and the possibility of it serving as an agent of zoonotic infection as inferred in a previous study¹¹ suggest the presence of a reservoir(s) for IOLA in the environment and a vector(s) to transmit it to humans.”

was changed to

“Considering the nature of IOLA as a **human-associated bacterium**, important issues to be addressed include the mechanisms and factors underlying the host-IOLA interaction. **For investing them, it is necessary to establish a cultivation method of IOLA.** In addition, the parasitic lifestyle of IOLA inferred in the current study and **its potential ability to inhabit animals other than human** as inferred in a previous study¹¹ suggest the presence of a reservoir(s) **and a vector(s)** for IOLA in the environment.” (Lines 393-398 in the revised manuscript).

3. I think it is likely that the long branch lengths in phylogenetic analysis do not suggest extremely rapid genetic evolution (lines 131 and 146) but instead reflect the fact that the genome is so divergent from the included references. Are there more closely-related references that can be included (eg the ones with ~73% homology, reported on line 111).

<Response>

We performed phylogenetic analyses (Fig. 3, Fig. 4, and Supplementary Fig. 4a and 5) using amino-acid sequences of house-keeping proteins of finished bacterial genomes, which were selected so as to cover a wide range of species. As shown in Fig. 2, IOLA protein sequences are highly divergent from the known proteins in the current database. In 16S rRNA sequence analyses (supplementary Table 4), we found uncultured bacterium clones showing 73% identities to the IOLA 16S rRNA as pointed by the reviewer, but only cloned 16S rRNA gene sequences are available. Although the long branch length is often used as a sign of rapid genetic evolution, we are not fully confident about the extremely rapid genetic evolution of IOLA. So, we have deleted “**suggesting extremely rapid genetic evolution**” (line 131 in the original manuscript), and changed “a rapidly evolving” (line 146 in the original manuscript) to “**a long branching**” (line 160 in the revised manuscript).

REVIEWERS' COMMENTS:

Reviewer #1 (Remarks to the Author):

The authors have sufficiently addressed my reviewer comments on the original submission with their revisions to the manuscript.

Reviewer #3 (Remarks to the Author):

The authors have appropriately responded to my prior comments, and in particular I think the interpretations have been nicely modified to indicate uncertainty that the newly-sequenced organism is a human pathogen.